# Effects of turning frequency on the nutrients of *Camellia oleifera* shell co-compost with goat dung and evaluation of co-compost maturity

**Jinping Zhang***, **Yue Ying, Xiaohua Yao**

Research Institue of subtropical Forestry, Chinese Academy of Forestry, Hangzhou, Zhejiang, China

* jinpingzhang@126.com

**Data Availability Statement:** All relevant data are within the manuscript and its Supporting Information files.

**Funding:** This work was financially supported by the Provincial Department of Science and

## Abstract

Composting is an important treatment method for *Camellia oleifera* shell and goat dung, which is crucial for the development of tea oil industry and goat breeding industry. Ventilation is the important regulatory factor in composting process, and high degree of maturity is the principal requirement for safe application of composting product. In the study, four treatments were designed as every 5 days turning (A1), every 7 days turning (A2), every 10 days turning(A3), and every 15 days turning (A4) for evaluating the maturity of *Camellia oleifera* shell co-composted with goat dung and optimizing turning frequency. During composting, TN, $NO_3^- -N$, GI, Solvita maturity index was increased along with composting process, while $NH_4^+ -N$, C/N shown an opposite trend. For all treatments, the longest thermophilic period (over 50°C), highest total nutrient and lowest C/N ratio were observed in A2. A turning frequency of every 7 days for co-compost of *Camellia oleifera* shell and goat dung could enhance the composting product quality. According to the analysis of spearman correlation, TN, C/N ratio, and GI could be used to comprehensively evaluate the compost maturity.

## Introduction

*Camellia oleifera* (*C. oleifera*) is one of the important tree species in China's non-wood product forests. *C. oleifera*, oil palm (*Elaeis guineensis* Jacq.), olive (*Olea europaea* L.), and coconut (*Cocos nucifera* L.) are the four major tree species in the world that produce woody edible oil [1]. *C. oleifera* shell is an important byproduct from the processing of *C. oleifera*, whose raw weight is approximately 1.5 times that of the *C. oleifera* seeds [2]. *C. oleifera* shell is rich in lignocellulose, saponins and tannins which were not conducive to microbial degradation and metabolism [3,4]. In addition, *C. oleifera* shell is rich in polysaccharides, proteins, fats, and other organic matter, as well as many trace elements and biologically active substances essential for plant growth, which make it a valuable biomass resource [5]. The output of *C. oleifera* seed reached more than 2.3 million tons by 2018, more than twice that of 2008, and a large number of *C. oleifera* shell was also produced [6,7]. Most of the shell is directly discarded for natural decomposition or burned as fuel. Waste gas, liquid, and residue produced in the treatment process pose potential threats to environmental safety [8].

Technology of Zhejiang, China, Grant NO.
2017C02022. The funder had no role in study
design, data collection and analysis, decision to
publish, or preparation of the manuscript.

**Competing interests:** The authors have declared
that no competing interests exist.

Composting is an effective means of processing *C. oleifera* shell and other solid organic waste [9]. However, given the physical and chemical properties of *C. oleifera* shell, particularly due to the presence of woody fibers, it is not suitable for self-composting [10]. Co-composting refers to the composting of raw materials together with other waste, such as dairy manure with straw [11], and chicken manure with sawdust [12]. Adding appropriate waste materials for co-composting can accelerate organic matter decomposition, shorten composting time, and enhance compost quality. Similar to pig and chicken manure, goat dung has high level of C, N, P, K and provides additional microorganisms for improved composting [13,14]. To accelerate the composting process of *C. oleifera* shell and improve compost quality, goat dung can be added for co-composting.

Through composting, organic chemicals in the shell are transformed by microbes into a stable, usable, and high value-added compost product, which helps prevent environmental pollution. Such products not only provide nutrients for plant growth but also improve soil properties, such as nutrients increase in soil and physical properties improvement of soil [15]. Furthermore, most of the harmful organisms (including pests, weed seeds, viruses and pathogens et al.) are killed during composting, which reduces the incidence of diseases [16].

Ventilation is an important controlling factor in the composting process. Appropriate ventilation can control the temperature of the compost pile, remove excess water and carbon dioxide, provide oxygen for microbial activity, and influence the physical and chemical properties of the final compost product [17]. Turning is a common method of altering the ventilation. High turning frequency will adversely decrease the temperature of the compost pile will, and will increase the composting time. Low turning frequency could result in anaerobic fermentation in the compost pile and production of ozone and other harmful substances [18]. Hence, an appropriate turning frequency is an important factor for enhancing the composting efficiency.

Maturity is an important criterion for evaluating the safety and stability of compost product. When microbial activity and plant toxin concentrations decrease to a safe level and compost no longer undergoes apparent material changes, the compost is considered to be 'mature'. One of the aims of choosing the appropriate turning frequency is to get the compost to be 'mature' quickly. Compost product not meeting maturity standards can easily cause environmental pollution [16]. However, due to the diversity and heterogeneity of raw materials, substantial variability exists in our understanding of the indices used to evaluate compost product maturity [19]. Bernal et al. (2009) [10] and Nolan et al. (2011) [20] noted that compost maturity could not be well described by any single attribute or parameter. Many studies used various physical, chemical (e.g., pH, temperature, and total organic carbon), and biological parameters [e.g., germination index (GI), seedling growth quality] to determine compost maturity [21–23]. In order to better judge the effect of turning frequency on compost maturity, it is very important to select appropriate parameters that can reflect the change of maturity.

In this study, *C. oleifera* shell and goat dung were used as the raw materials of compost. The temperature, pH, different nitrogen forms, GI and other parameters of the compost pile during composting were analyzed. Standard maturity indices suitable for the co-compost of *C. oleifera* shell with goat dung were determined. The turning frequency of compost was also optimized to provide a theoretical basis for the composting of *C. oleifera* shell.

## Materials and methods

### Descriptions of the study area

The composting experiment was performed during a 76-day period from October to December 2012. The study area was located in the Research Institute of Subtropical Forestry, Chinese

Academy of Forestry in Fuyang District, Hangzhou City, Zhejiang Province, China. Its altitude is 26.1 m, and its coordinates are approximately N 30˚3'31", E 119˚57'11" (latitude/longitude). During the study, the highest and lowest temperatures of that area were 24˚C and 3˚C, respectively, and the highest and lowest air humidity measurements were 84% and 32%, respectively.

*C. oleifera* shell used in the experiment was obtained from the East Red Forest Farm in Jinhua City, Zhejiang Province. Goat dung was obtained from a farm in Jilong Mountain, Fuyang District, Hangzhou city, Zhejiang Province. The effective microorganisms (EM) microbial agent was purchased from Henan Nanhua Qianmu Biotechnology Co., Ltd., and its main ingredients were *Bacillus*, *Lactobacillus*, *Bifidobacterium*, yeast, photosynthetic bacteria, acetic acid bacteria, *Actinobacillus*, and other original species.

## Experimental set-up

*C. oleifera* shell was crushed, passed through an 8-mm sieve, and co-composted with fresh goat dung. The dry mass ratio of the shell and goat dung was 4:1 in all experimental treatments. The EM microbial agent (weighing 3% of the dry mass of shell and goat dung) and urea were added to adjust the initial carbon-to-nitrogen ratio (C/N) to 30, and the initial moisture content to 55%. No adjustments of these parameters were made thereafter. In this experiment, box composting was used, which took place in an insulated and well-ventilated eco-composter (outer dimensions: 73 cm × 115 cm × 80 cm; volume: 220 L; manufacturer: Biolan). During the experiment, the cover of the composter was closed, and the ventilation valve was rotated to the maximum ventilation level. The properties of raw materials for composting (based on the determination of dry matter) are shown in Table 1.

There were 4 treatments (A1, A2, A3, and A4) in this experiment, and they referred to turning the compost once every 5, 7, 10, and 15 days, respectively. The properties of raw materials used in this study are shown in Table 1.

## Sample collection and analytical methods

During composting, the temperature of the compost pile (depth of the pile was 1.2m) was measured in the upper (10 cm below the top of the pile), middle, and lower (10 cm above the bottom of the pile) parts at approximately 3 pm every day. Ambient temperature was recorded at the same time. A five-point sampling method [24] was adopted to collect compost samples. A portion of the fresh samples were stored at -20˚C, and the remainder was dried at 65˚C. The determination of pH, total organic carbon (TOC), total nitrogen (TN), total phosphorus (TP), total potassium (TK), $NO_3^- - N$, and $NH_4^+ - N$ were based on the method used by Meng et al. (2018a, 2018b) [10,25]. GI was determined according to the method used by Zhao et al. (2013) [24]. Moisture content was obtained by drying at 105˚C for 24h in a hot-air oven. The Solvita maturity index was determined according to the "Guide to Solvita testing for compost maturity index" (Woods End Research, 2002).

**Table 1. The initial properties of compost materials[a].**

| Material | TOC(%) | TN(g/kg) | TP(g/kg) | TK(g/kg) | pH |
|---|---|---|---|---|---|
| Shell *of Camellia oleifera* | 48.6±1.8 | 4.19±0.01 | 0.17±0.03 | 8.54±0.11 | 5.54±0.02 |
| Goat dung | 16.0±2.1 | 0.55±0.03 | 0.22±0.07 | 0.53±0.06 | 8.32±0.01 |

Mean and standard error are shown (n = 3).

[a] Measured based on dry matter.

## Statistical analysis

Spearman correlational analysis was performed using SPSS 20.0 (IBM Co., Armonk, NY, USA). Principal Component Analysis (PCA) was performed using the Canoco 5.0 software package (Microcomputer Power, USA).

## Results and discussion

### Temperature

Temperature is an important indicator for monitoring the composting process. The optimum temperature range for composting is 40–65°C, and temperature exceeding 55°C is a necessary condition for killing pathogenic microbes, insect eggs, and weed seeds and for ensuring the compost product is harmless (Bernal et al., 2009). As shown in Fig 1, temperature in all treatments rapidly increased after the start of composting. A2 entered the thermophilic phase (>50°C) on 9th day, and A1, A3, and A4 entered the phase on the 12th, 32nd, and 11th day, respectively. The maximum temperatures of A1, A2, A3, and A4 were 72.7°C, 71.8°C, 62.7°C, and 68.67°C, respectively, which occurred on the 25th, 13th, 34th, and 19th day of the composting period, respectively. In all treatments, the thermophilic phase lasted for more than 4

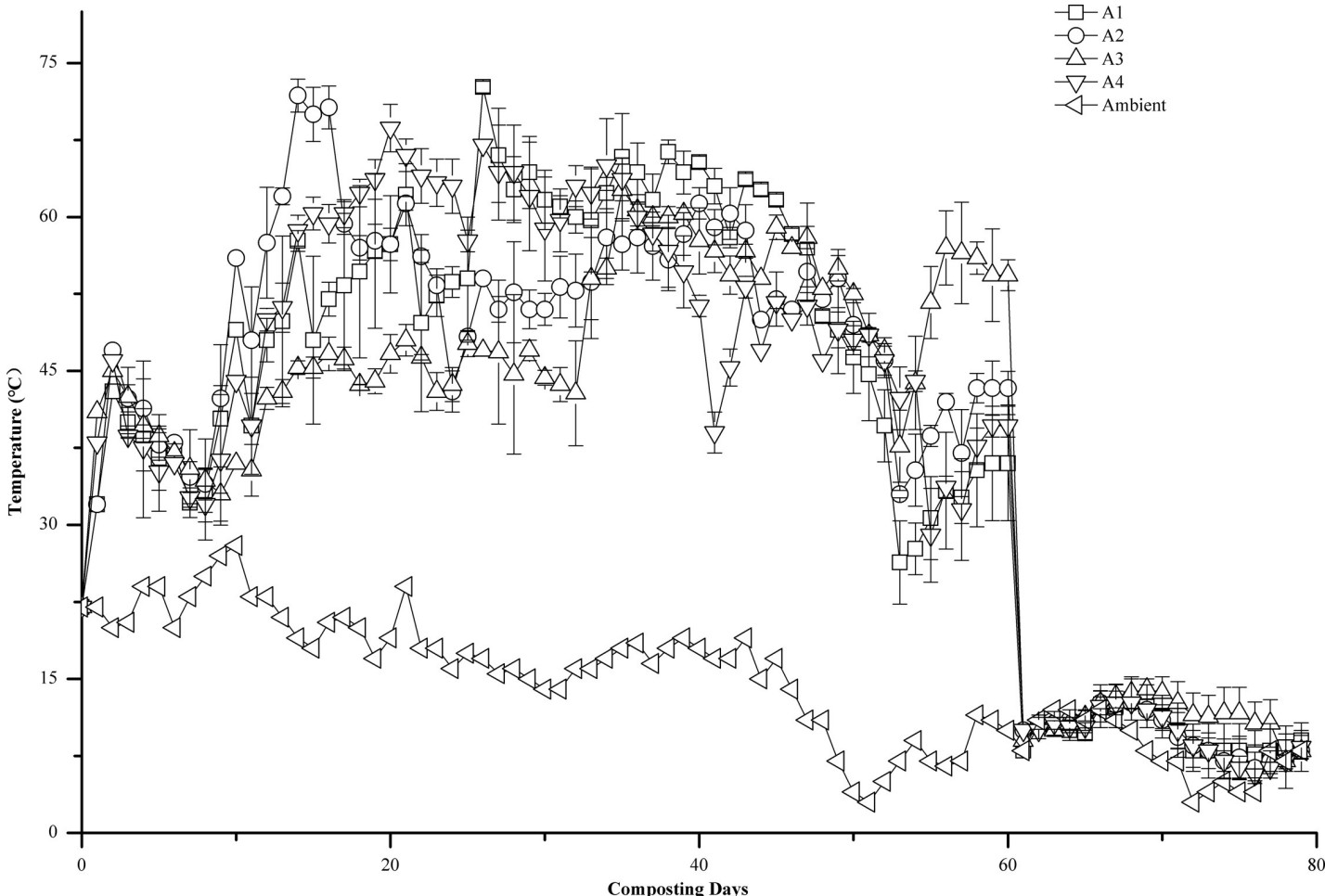

**Fig 1. Temperature profile of the composting over time.** Error bars represent the standard deviations of the means (n = 3).

consecutive weeks, particularly in A2, where it lasted for 41 consecutive days, and all treatments met hygiene and safety standards (temperature above 55°C for more than 3 consecutive days) (GB 7959–2012, China). After approximately 50 days of composting, the thermophilic phase ended, and the temperature of the compost pile decreased gradually. Between 50 and 60 days after composting, due to reduced temperature and the accumulation of soluble organic matter, microbial activity increased again [10], leading to another stage of temperature rise in the compost pile, and after 60 days of composting, the temperature in all treatments rapidly decreased to ambient temperature. This may be due to a lot of heat lost by turning over the heap, far more than the heat generated by the material metabolism of microorganisms. In terms of temperature variability, A2 had a faster rate of temperature increase and the longest thermophilic phase.

## pH

The pH is one of the important factors affecting the growth and reproduction of microbes. Microbial activity can be inhibited at too high or too low of pH, which hinders composting [18]. As shown in Fig 2, the initial pH in A1, A2, A3, and A4 was 6.29±0.11, 6.32±01, 6.27 ±0.04, and 6.24±0.07, respectively. The optimum pH for composting is 5.5–8.0 [26,27]. During the first 20 days of composting, the pH in all treatments increased rapidly. On the 20th day of

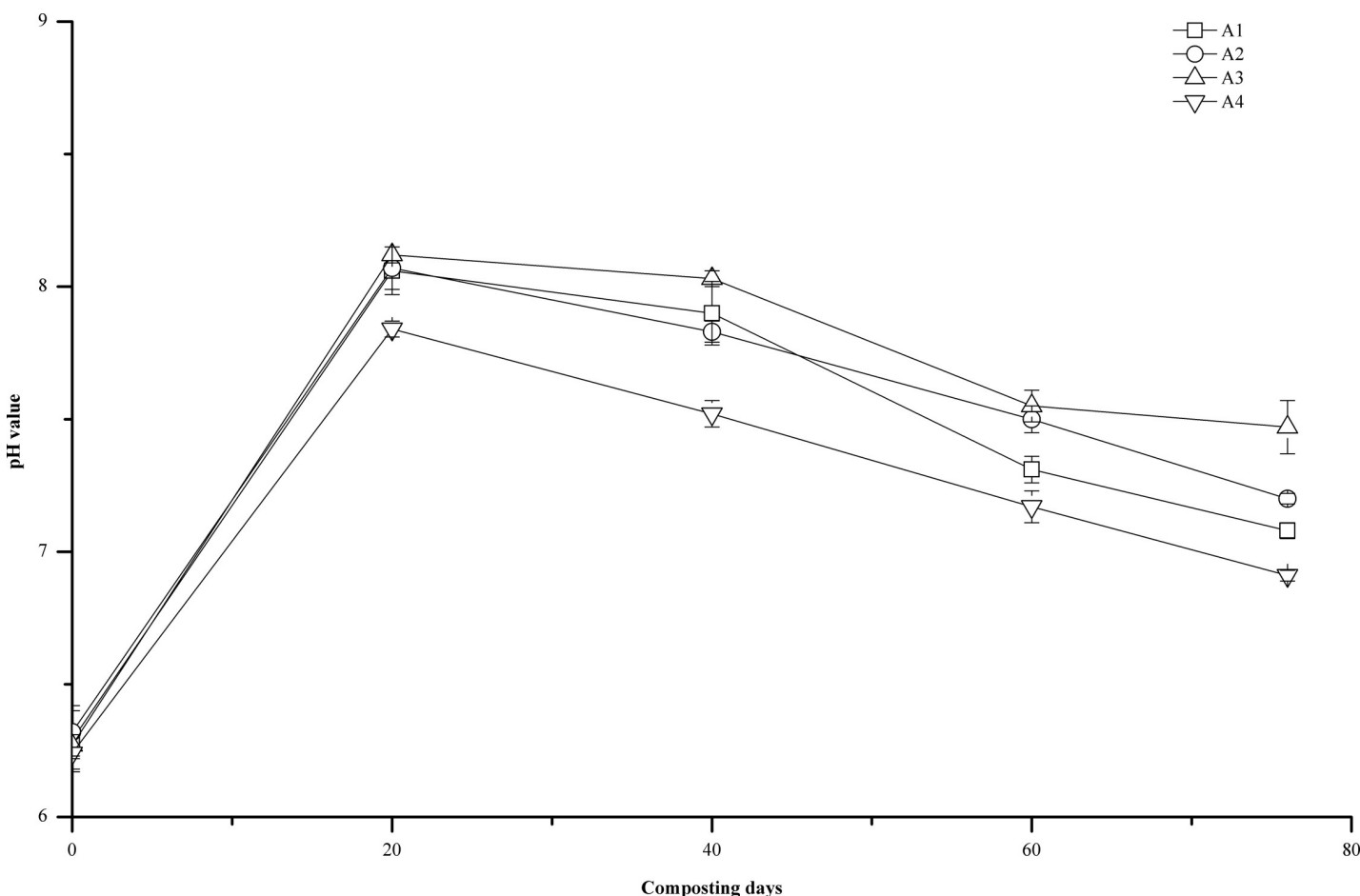

**Fig 2. Changes of pH during composting at different turning frequencies.** Error bars represent the standard deviations of the means (n = 3).

the composting period, the pH in A1, A2, A3, and A4 was 8.06±0.09, 8.07±0.08, 8.12±0.03, and 7.84±0.03, respectively. This pH might be due to the breakdown of phenolic acid compounds in the raw materials, as well as the breakdown of proteins and amino acids by microbes that produce ammonia and other alkaline substances [11]. The pH of the compost pile decreased afterwards, which might be caused by the formation of organic acids and phenolic compounds [28,29]. At the end of composting, the pH of the composted product in A1, A2, A3, and A4 was 7.08±0.03, 7.21±0.02, 7.47±0.10, and 6.91±0.02, respectively, which are all in the suitable range of seedling growth (Bustamante et al. 2008).

## Total nitrogen

Nitrogen is the main elemental nutrient in the compost product. Improper composting procedures can result in a substantial loss of nitrogen. Not only does the compost product lacking nitrogen have low fertilizer efficiency, it may take up soil nitrogen, thereby resulting in a soil nutrient imbalance [17]. During the 76-day composting, the TN content in all treatments increased variably (Fig 3) and was significantly positively correlated with the number of days of composting ($r = 0.952$, $P<0.05$; $r = 0.900$, $P<0.05$; $r = 0.976$, $P<0.01$; $r = 0.965$, $P<0.001$). The TN content in A1, A2, A3, and A4 increased by 88.9%, 99.8%, 83.9%, and 103.4%,

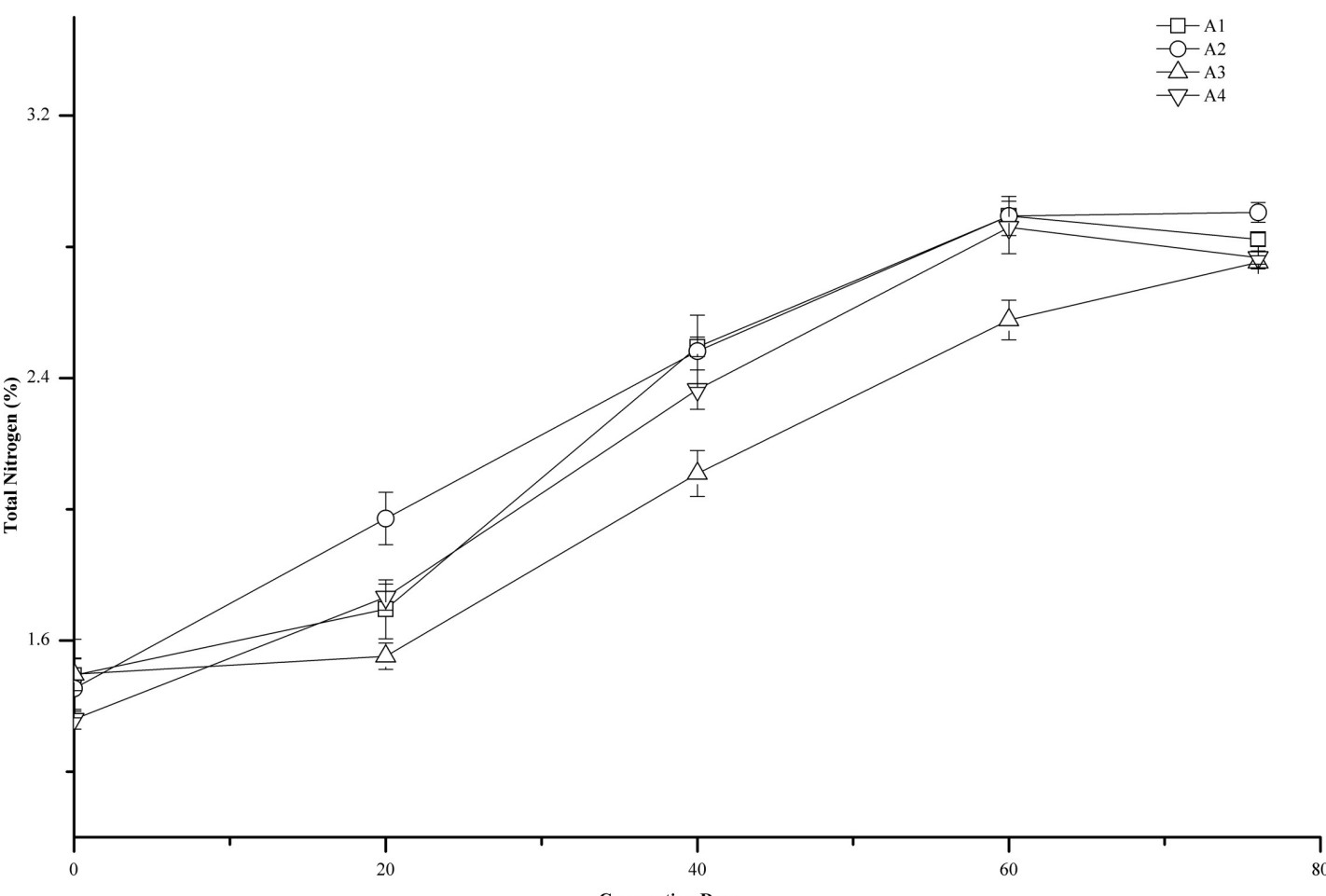

**Fig 3. Changes of total nitrogen content (%) during composting at different turning frequencies.** Error bars represent the standard deviations of the means (n = 3).

respectively. This might be caused by the mineralization of organic matter during composting, loss of $CO_2$, and evaporative loss of water driven by heat production during the oxidation of organic matter [30].

## $NO_3^- - N$ and $NH_4^+ - N$

$NO_3^- - N$ is the main source of nitrogen for most plants [31]. The higher the $NO_3^- - N$ content, the higher the fertilizer efficiency of the compost product. As shown in Fig 4, between 0 and 40 days of composting, $NO_3^- - N$ did not change significantly in all treatments. After 40 days of composting, the $NO_3^- - N$ increased rapidly. In contrast, the $NH_4^+ - N$ content exhibited a decreasing trend after the start of composting. This is because during composting, the proteins, amino acids and other organic matter were first utilized by microbes as energy and nitrogen sources for them, and when these substances were utilized, the $NH_4^+ - N$ formed during microbial deamination was unstable, part of which was volatilized as ammonia gas, and part of which was converted to $NO_3^- - N$ by ammonia-oxidizing microbes [32]. The former process mainly occurred in the early stage of composting when pH and temperature were relatively high, and ammonia volatilization caused the rapid reduction of $NH_4^+ - N$ in the compost pile.

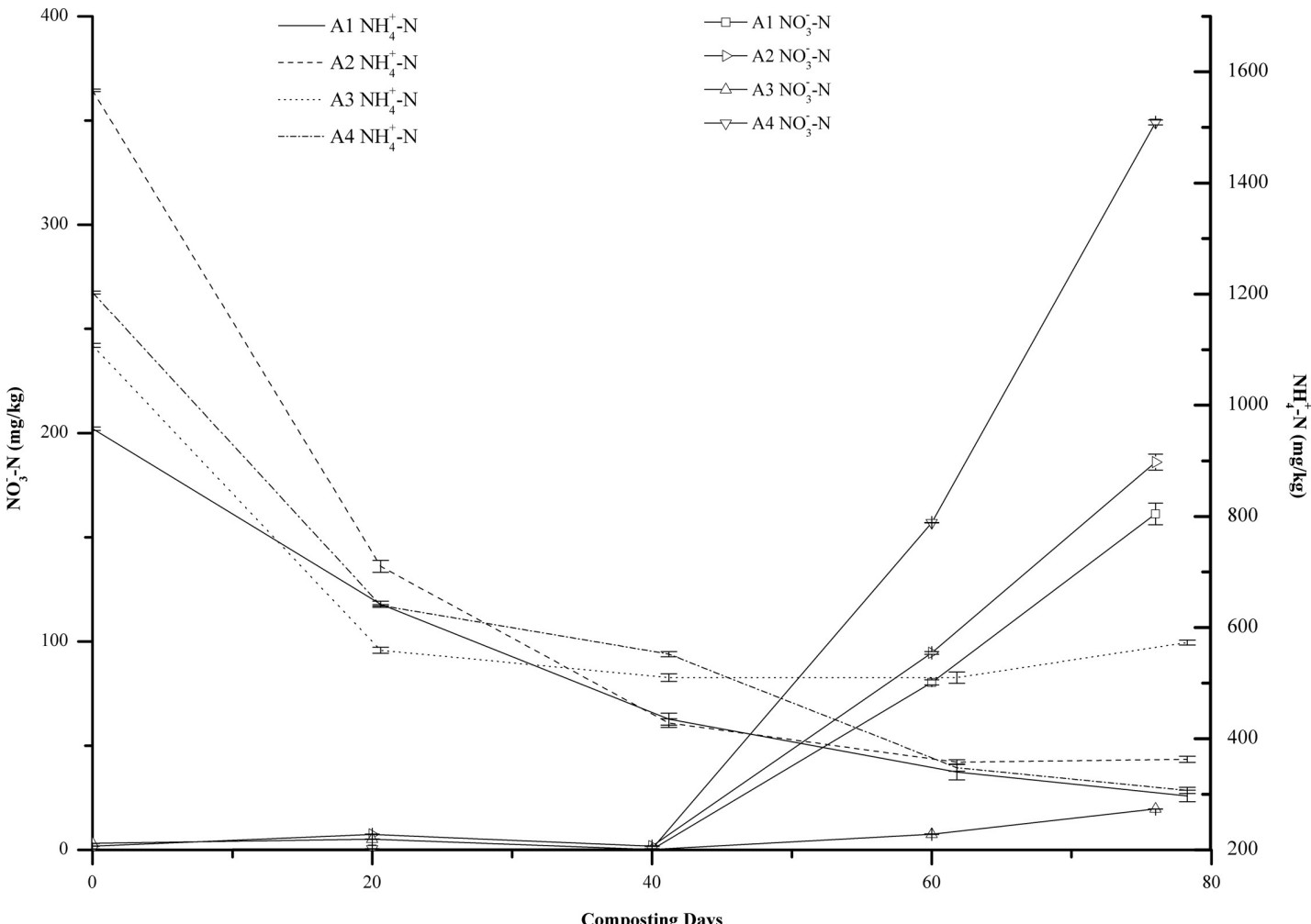

**Fig 4. Content (%) of $NO_3^- - N$ and $NH_4^+ - N$ during composting at different turning frequencies.** Error bars represent the standard deviations of the means (n = 3).

The latter process mainly occurred in the mid to late stages of composting and was the main reason of $NO_3^- - N$ increase. This is because nitrifying bacteria cannot grow at temperatures above 40°C, and hence, nitrification mainly occurred during the stages of temperature decrease and compost maturity [25]. At the end of composting, the magnitude of decrease in the $NH_4^+ - N$ far exceeded the magnitude of increase in the $NO_3^- - N$, which might be due to the substantial volatilization of $NH_4^+ - N$ at high temperatures [26].

## C/N ratio

As shown in Fig 5, during composting, the C/N ratio exhibited an apparent decreasing trend in all treatments. The results of the Pearson correlation analysis indicated that the C/N was significantly negatively correlated with the number of days of composting (r = -0.939, r = -0.951, r = -0.944, r = -0.934, P<0.05, respectively). Dalal et al. showed that, normally when C/N of organic fertilizers was higher than 30, the biological immobilization effects of the mineral nitrogen in the soil were greater than the mineralization effects of organic nitrogen. When C/N was between 20–30, the rate of biological immobilization of mineral nitrogen was similar to the rate of mineralization of organic nitrogen, and the available nitrogen in the soil did not

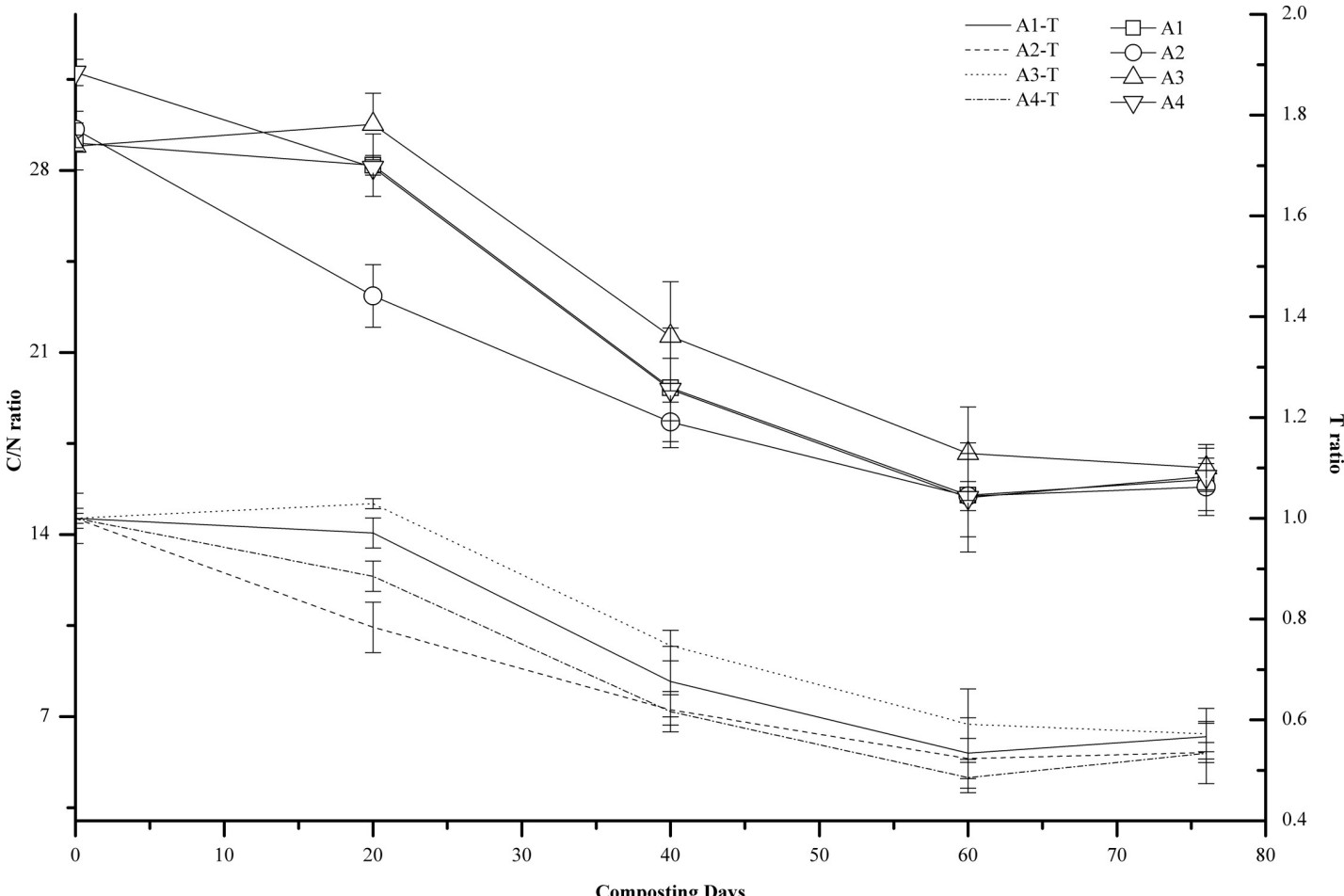

**Fig 5. Changes of C/N ratio and T ratio during composting at different turning frequencies.** Error bars represent the standard deviations of the means (n = 3). T = (C/N $_{final}$)/(C/N $_{initial}$).

change significantly. When C/N was below 20, the rate of mineralization of organic nitrogen was greater than the rate of biological immobilization of mineral nitrogen, and the available nitrogen in the soil increased. It is generally believed that the lower C/N ratio, the better the quality of organic fertilizer [33]. After 76 days of composting, the C/N of the compost product in A1, A2, A3, and A4 was 16.12±1.21, 15.83±1.10, 16.56±0.90, and 16.22±0.51, respectively, which led to the increase in available nitrogen in the soil.

According to the methods of the US Composting Council (2002), the compost is considered mature when its C/N is less than 21. Itavaara et al. (1997) [34] and Arja et al. (1997) [35] suggested that when T<0.6 [T = (C/N) $_{final}$/(C/N) $_{initial}$], the compost had reached maturity. Based on the C/N ratio, the compost in A1, A2, and A4 reached maturity on the 60th day of composting. Based on the T value, the compost in A1, A2, and A4 reached maturity on the 60th day of composting, whereas A3 reached maturity on the 76th day of composting.

## Total nutrient content

The total nutrient content (%) refers to the sum of the N, P (calculated as $P_2O_5$), and K (calculated as $K_2O$) contents. N, P, and K are all essential elemental nutrients for plant growth, and total nutrient content (%) greater than 5% is an important indicator of a compost product that is being considered an organic fertilizer [Chinese standard for organic fertilizers (NY525-2012)]. As shown in Fig 6, during composting, the total nutrient content in all treatments was significantly positively correlated with the number of days of composting (r = 0.974, r = 0.972, r = 0.976, r = 0.995, P<0.01, respectively). The total nutrient contents of A2, A3, and A4 were greater than 5% on the 60th day, and that in A1 was greater than 5% on the 76th day. When the composting ended (76th day), the total nutrient content in A1, A2, A3, and A4 was 5.41 ±0.14%, 5.75±0.11%, 5.62±0.12%, and 5.53±0.06%, respectively. Among these treatments, the total nutrient content was the highest in A2, indicating that too high or too low of turning frequency could cause the loss of total nutrients. In this experiment, the total nutrient content in the compost product was the highest when the turning frequency was once every 7 days.

## Germination index (GI)

The GI can effectively reflect the phytotoxicity of the compost. The compost is considered basically nontoxic when the GI is greater than 60%, and the phytotoxicity of the compost completely disappears when the GI is greater than 80% [31]. As shown in Fig 7, the GI of Bok choy (*Brassica chinensis* L.) in the compost pile exhibited an increasing trend in the different treatments and in the different composting stages. In particular, GI in the compost pile in A1 and A3 decreased considerably after 14–20 days of composting. This might be related to the phytotoxicity of small molecules in the compost, such as ammonia, phenols, and acids [18]. At the end of composting, the GI in A1, A2, A3, and A4 was 110.12%, 103.00%, 73.94%, and 108.41%, respectively. GI in A1, A2, and A4 was 48.93%, 39.30%, and 46.62% higher than that in A3, respectively. This result indicated that the final compost in A1, A2, and A4 did not have phytotoxicity [19], and the final compost in A3 was still toxic to plants. Therefore, turning could accelerate the breakdown and conversion of phytotoxic substances in the compost pile and facilitate the maturation of compost. when the turning frequency is low enough, sufficient anaerobic fermentation can also reduce the phytotoxicity of the compost [36]. However, GI was measured by using leachate from compost, which cannot explain the harmful effects of many insoluble substances in compost products on plant growth, nor can explain the effects on plant root growth and aboveground development. Therefore, GI alone cannot prove that compost products are maturity [37].

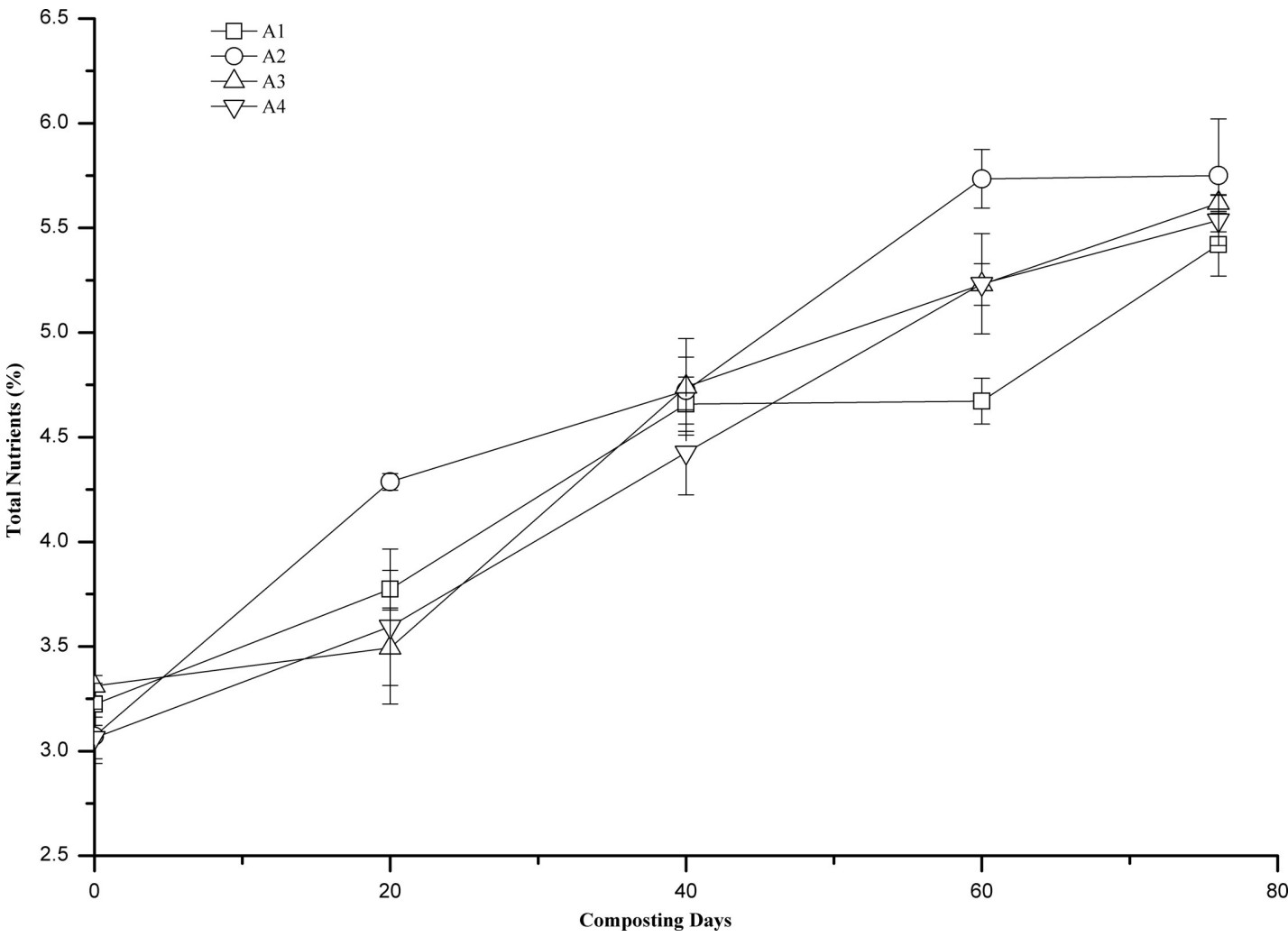

**Fig 6. Changes of total Nutrition (%) during composting at different turning frequencies.** Error bars represent the standard deviations of the means (n = 3).

### Solvita maturity index

Solvita maturity index is a widely accepted index obtained through simple tests [38]. As shown in Table 2, after the start of composting, The Solvita maturity index increased gradually in all treatments. At the end of composting, the Solvita maturity index in A1, A2, A3, and A4 was 7, 7, 5, and 7 respectively. When the Solvita maturity index is 7, composting is considered "finished," according to the "Guide to Solvita testing for compost maturity index" (Woods End Research, 2002). Therefore, the compost product in A1, A2, and A4 could be regarded as mature after 76 days of composting (Table 2).

### Evaluation system of maturity indices

Many parameters have been used as indicators for evaluating compost maturity [21,22]. In this study, the Solvita maturity index was used as a standard index, and its correlations with parameters such as TN, C/N, and GI were analyzed. As shown in Table 3, both TN, C/N, and GI in four treatments were significantly correlated with the Solvita maturity index (P<0.05). The C/N was negatively correlated with other maturity parameters (TN, GI, and Solvita maturity

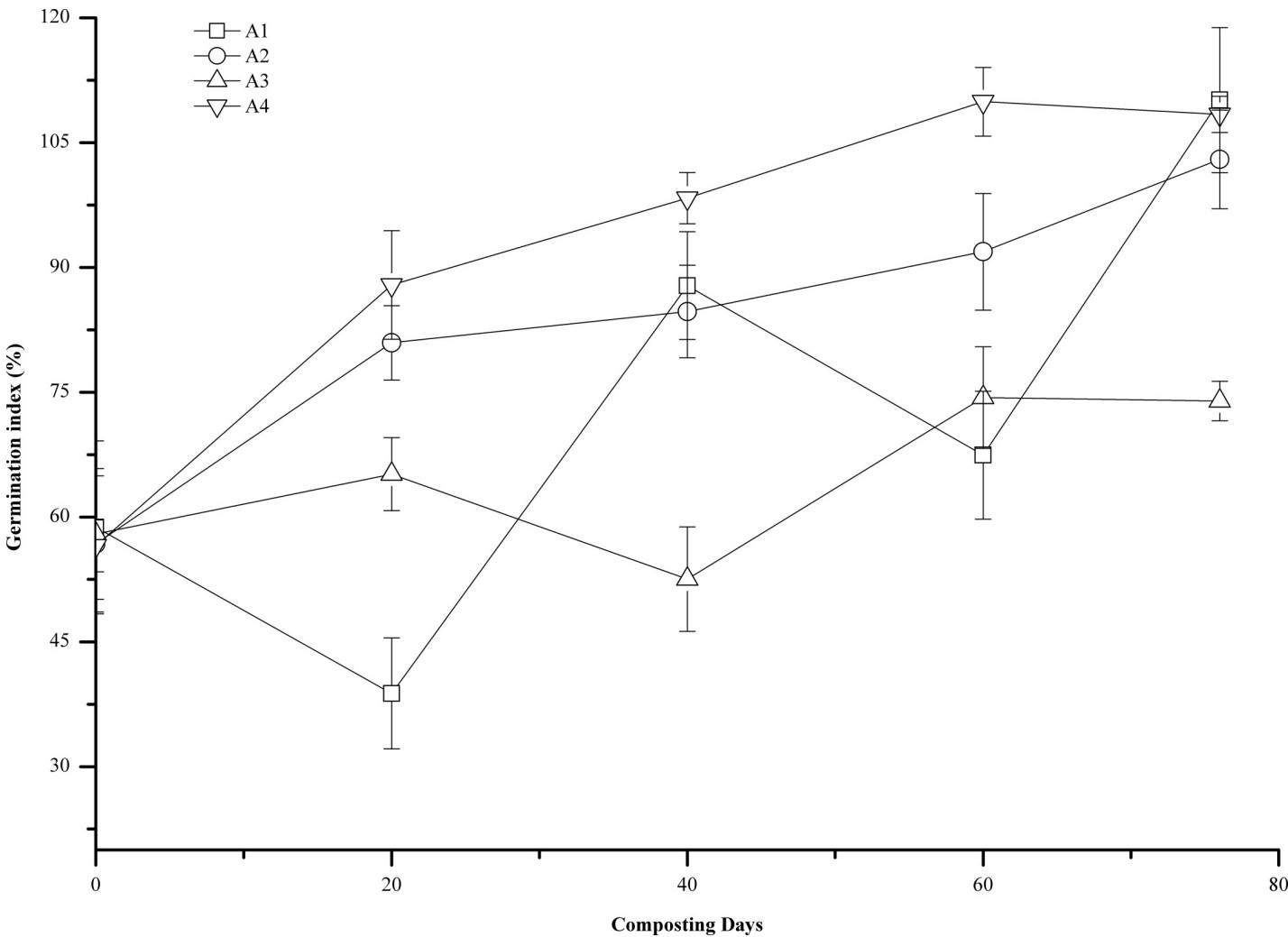

**Fig 7. Changes of Germination index(%)during composting at different turning frequencies.** Error bars represent the standard deviations of the means (n = 3).

index). The TN, GI, and Solvita maturity index were positively correlated with each other. During composting, the TN, GI, and Solvita maturity index showed an increasing trend, and the C/N showed a decreasing trend. A principal component analysis (PCA) was used to identify factors that could distinguish the relationships between physicochemical parameters (Fig 8). The parameters were grouped according to principal components in four clearly distinguished clusters. The first group was $NO_3^- - N$ and Solvita maturity index, the second group was $NH_4^+ - N$, C/N, the third group was TN and Total nutrient, and the fourth group was pH.

**Table 2. Solvita maturity index during composting.**

| Composting Days | A1 | A2 | A3 | A4 |
|---|---|---|---|---|
| 0 | N.D. | N.D. | N.D. | N.D. |
| 20 | 2 | 1 | 4 | 3 |
| 40 | 5 | 5 | 4 | 4 |
| 60 | 6 | 6 | 5 | 6 |
| 76 | 7 | 7 | 5 | 7 |

**Table 3. Bivariate (Pearson) correlation analysis of the physical and chemical parameters of each group.**

|  | Parameter | TN(%) | C/N | GI(%) | SOLVITA |
|---|---|---|---|---|---|
| A1 | TN(%) | 1 | -0.998** | 0.835 | 0.953* |
|  | C/N |  | 1 | -0.851 | -0.969** |
|  | GI(%) |  |  | 1 | 0.914* |
|  | SOLVITA |  |  |  | 1 |
| A2 | TN(%) | 1 | -0.944* | 0.975** | 0.934* |
|  | C/N |  | 1 | -0.746 | -0.619 |
|  | GI(%) |  |  | 1 | 0.981** |
|  | SOLVITA |  |  |  | 1 |
| A3 | TN(%) | 1 | -0.989** | 0.647 | 0.902* |
|  | C/N |  | 1 | -0.561 | -0.864 |
|  | GI(%) |  |  | 1 | 0.886* |
|  | SOLVITA |  |  |  | 1 |
| A4 | TN(%) | 1 | -0.995** | 0.976** | 0.948* |
|  | C/N |  | 1 | -0.973** | -0.917* |
|  | GI(%) |  |  | 1 | 0.892* |
|  | SOLVITA |  |  |  | 1 |

**. Correlation at the significant level of 0.01.

*. Correlation at significant level of 0.05.

Therefore, the inclusion of TN, C/N and GI in the evaluation system of compost maturity can allow compost maturity to be effectively evaluated.

By studying the changes in compost maturity under 4 different turning frequencies, observations on the co-compost of *C. oleifera* shell with goat dung showed that on reaching maturity, the C/N ratio of the co-compost was between 15.83 and 16.22; The GI was greater than 100%, and the TN was between 2.85 and 2.91%.

## Final compost quality

The main chemical properties of the final compost products in the four treatments are shown in Table 4. The pH of the compost in each treatment was between 6.91 and 7.48, and its moisture content was below 30%. The highest total nutrient content of the compost product was 5.75±0.11% (A2), and the lowest was 5.41±0.14% (A1); both were greater than 5% and within the limits defined by the Chinese standard for organic fertilizers (NY525-2012). The organic matter content of the compost product in A2 was as high as 79.54±0.98%, and it was the lowest in A1 at 78.44±0.38%. These values were much higher than the requirement (45%) in the Chinese standard for organic fertilizers (NY525-2012).

## Conclusion

During the study, the products obtained from composting of *C. oleifera* shell and goat dung for 76 days were in accordance with the Chinese organic fertilizer standard NY525-2012. In the composting process, TN, $NO_3^- -N$, GI, Solvita maturity index increased with the composting process, while $NH_4^+ -N$, C/N showed the opposite trend. According to the correlation analysis, TN, C/N ratio and GI can be used to evaluate the maturity of compost of *C. oleifera* shell and goat dung comprehensively. Among the four treatments, the longest duration of thermophilic phase was observed in A2 for 42 days (>50°C), with the highest total nutrient (5.75 ±0.11%) and lowest C/N value (15.83±1.10). It can be seen that the turning frequency of every

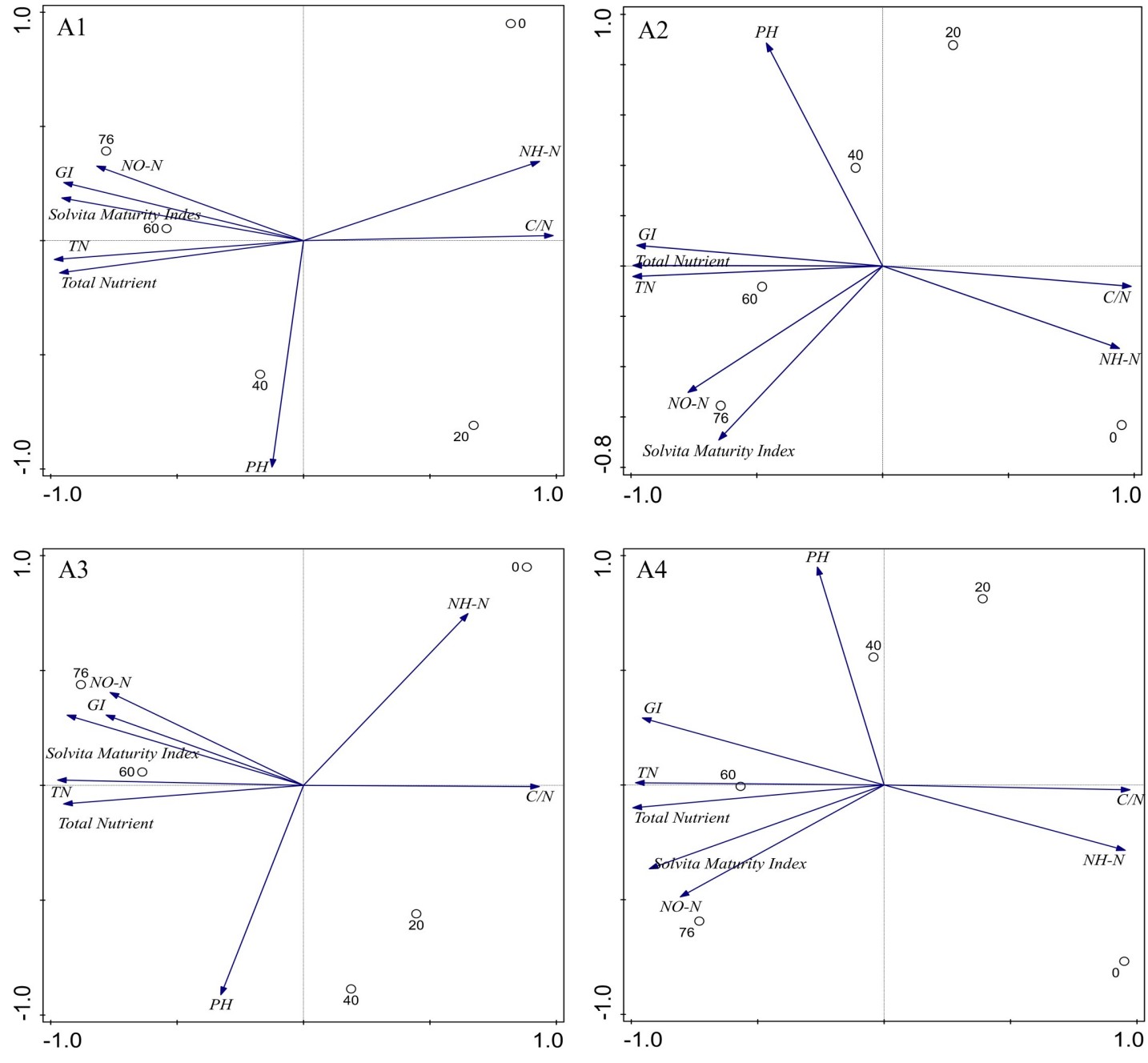

**Fig 8. PCA analysis of the parameters in composting process.**

**Table 4. Chemical characteristics of the final compost obtained.**

| Parameter | A1 | A2 | A3 | A4 | NY525-2012 |
|---|---|---|---|---|---|
| Moisture (%) | 15.0±0.4 | 15.1±0.1 | 14.3±0.6 | 14.8±0.3 | ≤30 |
| Organic matter content (%) | 78.44±0.38 | 79.54±0.98 | 78.61±0.56 | 77.41±1.02 | ≥45 |
| Total nutrients (%) | 5.41±0.14 | 5.75±0.11 | 5.62±0.12 | 5.53±0.06 | 5 |
| pH | 7.08±0.03 | 7.21±0.02 | 7.47±0.10 | 6.91±0.02 | 5.5–8.5 |

Except for moisture, all data are expressed on a dry weight basis.

7 days can improve the quality of compost products. However, due to the diversity of raw materials and composting conditions, we need to do more research to acquire parameters with simple determination method and strong correlation with maturity to determine the appropriate turning frequency for composting.

## Supporting information

**S1 File. Data of sample parameters during composting**
(XLSX)

## Author Contributions

**Conceptualization:** Jinping Zhang, Xiaohua Yao.

**Data curation:** Jinping Zhang.

**Formal analysis:** Jinping Zhang, Yue Ying.

**Investigation:** Jinping Zhang, Yue Ying, Xiaohua Yao.

**Methodology:** Jinping Zhang, Xiaohua Yao.

**Project administration:** Jinping Zhang.

**Resources:** Jinping Zhang, Xiaohua Yao.

**Software:** Yue Ying.

**Supervision:** Jinping Zhang.

**Writing – original draft:** Jinping Zhang, Yue Ying.

**Writing – review & editing:** Jinping Zhang, Yue Ying.

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
