## [Decision Letter · Decision Letter 0]

1 Aug 2019

PONE-D-19-19121

Effects of turning frequency on the nutrients of Camellia oleifera shell co-compost with sheep manure and evaluation of co-compost maturity

PLOS ONE

Dear Dr. Jinping Zhang,

Thank you for submitting your manuscript to PLOS ONE. After careful consideration, we feel that it has merit but does not fully meet PLOS ONE’s publication criteria as it currently stands. Therefore, we invite you to submit a revised version of the manuscript that addresses the points raised during the review process.

We would appreciate receiving your revised manuscript by September 10. To enhance the reproducibility of your results, we recommend that if applicable you deposit your laboratory protocols in protocols.io, where a protocol can be assigned its own identifier (DOI) such that it can be cited independently in the future. For instructions see: http://journals.plos.org/plosone/s/submission-guidelines#loc-laboratory-protocols

We look forward to receiving your revised manuscript.

Kind regards,

Sartaj Ahmad Bhat, Ph.D

Academic Editor

PLOS ONE

Journal Requirements:

Reviewers' comments:

Reviewer's Responses to Questions

**Comments to the Author**

1. Is the manuscript technically sound, and do the data support the conclusions?

Reviewer #1: Yes

Reviewer #2: Yes

2. Has the statistical analysis been performed appropriately and rigorously? 

Reviewer #1: No

Reviewer #2: Yes

3. Have the authors made all data underlying the findings in their manuscript fully available?

Reviewer #1: No

Reviewer #2: Yes

4. Is the manuscript presented in an intelligible fashion and written in standard English?

Reviewer #1: No

Reviewer #2: Yes

5. Review Comments to the Author

Reviewer #1: The topic of the manuscript is actual. More details are necessary in the Introduction section of the manuscript. Some references need to be included. Some section needs to be rewritten.

Specific comments are listed below.

Abstract

Why it is so important to undertake these trials？What’s the purpose?

Why shell? Why manure? Why combined?

Line 12, “could enhance” compare with what?

Sample conclusion is missing.

Introduction

Lines 26-28, please rephrase these sentences.

Line 27, “slowly”, how long?

Lines 28-30, reference(s)?

Line 31, amount of the production? Add reference(s).

Lines 42-45, reference(s)? Why sheep manure?

Line 46, “improve”, how?

Line 47, “pests”, for examples. Pests, including viruses and pathogens? What’s the meaning?

Lines 53-56,”high” and “low” are generic terms. Please indicate a quantity.

Lines 59-70, this subsection needs to be rewritten. What’s the key point for this paragraph?

Lines 72-73, the purpose for testing these parameters.

Line 94, water content of the shell.

Line 95, 55% is water content or relative humidity?

Lines 113-115, depth of the pile.

Lines 124-125, more details.

Results and discussion

It’s important to add statistical analysis of the difference between different treatments.

Lines 133-135, please rephrase these sentences.

Line 140, (GB 7959-87), China?

Lines 175-176, 211, 241-242, r and P, in italics.

Conclusion

Please re-write the conclusion. Based on the purpose and the actual data.

Reviewer #2: This study conducted a 76-day composting experiment to explore the effects of turning frequency on the nutrients of Camellia oleifera shell co-compost with sheep manure and evaluation of co-compost maturity. The results showed that turning once every 7 days is the most suitable way to enhance the composting efficiency and compost product quality. In addition, this study proposed that TN, carbon-to-nitrogen ratio (C/N), and GI are useful indexes for comprehensively evaluate the compost maturity. In general, this work has some significance for practical production. However, further analysis and discussion about the relationships among these indexes are needed in this manuscript using a better statistical analysis method. Moreover, there some small problems needed to be revised. Thus, I think the present paper need a revision before publication. I hope my comments would be of help to the future improvement of the work.

1. Lines 284-294: Composting is a complex environmental process, many parameters will affect each other. Thus, I think Pearson correlation analysis is not sufficient to determine which factors are suitable for evaluating compost maturity. Multivariate analysis may be helpful to further understand the complex relationships among various parameters of composts. Redundancy analysis (RDA), structural equation models (SEMs) et al. may be a good choice, and the following references may helpful: Bioresource Technology 285 (2019) 121326; Environmental Pollution 250 (2019) 166-174.

2. The discussion section needs to be improved, and appropriate outlook should be supplemented.

3. Are “sheep manure” and “goat dung” the same in meaning? If they are same, please unify the words.

4. The method of moisture determination should be introduced in the “materials and methods” section.

5. Lines 145-146: why the temperature decreased to room temperature suddenly in all treatments?

6. The resolution of the pictures in the manuscript is poor.

6. PLOS authors have the option to publish the peer review history of their article (what does this mean?). If published, this will include your full peer review and any attached files.

Reviewer #1: No

Reviewer #2: No

---

## [Author Response · Author response to Decision Letter 0]

15 Aug 2019

Reviewer #1: 

Abstract

Why it is so important to undertake these trials？What’s the purpose?

Why shell? Why manure? Why combined?

Line 12, “could enhance” compare with what?

Sample conclusion is missing.

The purpose and background of the experiment are supplemented in the abstract, but the reasons for the use of co-composting are explained in detail in the introduction. "Can enhance" is a comparison between different groups under different heap turnover frequencies.

Introduction

Lines 26-28, please rephrase these sentences.

Modified to " C. oleifera shell is rich in lignocellulose, saponins and tannins which were not conducive to microbial degradation and metabolism ".

Line 27, “slowly”, how long?

The standard of natural decay of C. oleifera shell shell is generally more than 6 months. Because there is no specific data, it is revised in this paper.

Lines 28-30, reference(s)?

Relevant references have been supplemented in this paper.

Line 31, amount of the production? Add reference(s).

It has been supplemented in the paper according to the references.

Lines 42-45, reference(s)? Why sheep manure?

Co-composting of livestock manure and other materials to make the composting process go smoothly is a common waste composting method, which has been used in many literatures. Similar to pig manure and chicken manure, sheep manure has a high content of CNPK, and can provide additional microorganisms for improving composting. It is also one of the common wastes around the experimental area. Therefore, sheep manure and Camellia shell are selected for co-composting.

Line 46, “improve”, how?

It has been supplemented in the article, “such as nutrients increase in soil and physical properties improvement of soil”.

Line 47, “pests”, for examples. Pests, including viruses and pathogens? What’s the meaning?

The pest means Harmful organisms, But there are misunderstandings in this paper. So we revised the expression in this paper. “Furthermore, most of the harmful organisms (including pests, weed seeds, viruses and pathogens et al.) are killed during composting, which reduces the incidence of diseases”

Lines 53-56,”high” and “low” are generic terms. Please indicate a quantity.

"High" and "low" are the relative values of overturning frequency, which can not give specific values. In the article, "high" refers to turning frequency of 5 days, while "low" refers to turning frequency of 15 days.

Lines 59-70, this subsection needs to be rewritten. What’s the key point for this paragraph?

This section is to illustrate the importance of compost maturity, because the compost product reaches maturity, so that the product can be used in production premise. Choosing the turning frequency is also to make the compost material ripen as quickly as possible. It also provides a reference for judging the availability of compost products and adjusting the turning frequency in practical operation.

Lines 72-73, the purpose for testing these parameters.

The purpose of testing these parameters is to find a simple and low-cost way to judge the maturity of compost products and to determine which turning frequency is more suitable for composting. Although Solvita index is internationally recognized as an effective index for judging maturity, its domestic price is still on the high side.

Line 94, water content of the shell.

This step is to control the size of Camellia shell. I think the moisture content of Camellia shell has nothing to do with this operation. The moisture content of Camellia shell will change during storage, so the moisture content of raw materials is not important and will not be explained. And in the follow-up operation, we use dry matter as the adjustment standard, the moisture content of raw materials in this step has no effect on the follow-up.

Line 95, 55% is water content or relative humidity?

55% is water content, and moisture content is also used in other composting literature. (Wang, Y. Q., Zhang, J. L., Schuchardt, F., & Wang, Y. (2014). Degradation of morphine in opium poppy processing waste composting. Bioresource Technology, 168(3), 235-239.)

Lines 113-115, depth of the pile.

It has been supplemented in this article. depth of the pile was 1.2m 

Lines 124-125, more details.

In this paper, the analysis method and the information of using software are supplemented（ Canoco 5.0 and PCA）.

Results and discussion

It’s important to add statistical analysis of the difference between different treatments.

PCA analysis is supplemented to judge that the selected parameters are reasonable and representative.

Lines 133-135, please rephrase these sentences.

These sentences have been amended to read "A2 enter the thermophilic phase (> 50 C) on 9th day, and A1, A3, and A4 enter the phase on the 12th, 32nd, and 11th day, respectively."

Line 140, (GB 7959-87), China?

Yes, this is China's national standard, which has been replaced. The standard is now changed to GB 7959-2012. China.

Lines 175-176, 211, 241-242, r and P, in italics.

I consulted the literature of many magazines and found no need to change it into italics. Therefore, there is no modification in the paper.

Conclusion

Please re-write the conclusion. Based on the purpose and the actual data.

The conclusion has been revised.

“During the study, the products obtained from composting of C. oleifera shell and goat dung for 76 days were in accordance with the Chinese organic fertilizer standard NY525-2012. In the composting process, TN, NO- 3-N, GI, Solvita maturity index increased with the composting process, while NH+ 4-N, C/N showed the opposite trend. According to the correlation analysis, TN, C/N ratio and GI can be used to evaluate the maturity of compost of C. oleifera shell and goat dung comprehensively. Among the four treatments, the longest duration of thermophilic phase was observed in A2 for 42 days (>50℃), with the highest total nutrient (5.75±0.11%) and lowest C/N value (15.83±1.10). It can be seen that the turning frequency of every 7 days can improve the quality of compost products. However, due to the diversity of raw materials and composting conditions, we need to do more research to acquire parameters with simple determination method and strong correlation with maturity to determine the appropriate turning frequency for composting.“

Reviewer #2: 

1. Lines 284-294: Composting is a complex environmental process, many parameters will affect each other. Thus, I think Pearson correlation analysis is not sufficient to determine which factors are suitable for evaluating compost maturity. Multivariate analysis may be helpful to further understand the complex relationships among various parameters of composts. Redundancy analysis (RDA), structural equation models (SEMs) et al. may be a good choice, and the following references may helpful: Bioresource Technology 285 (2019) 121326; Environmental Pollution 250 (2019) 166-174.

PCA analysis is supplemented to judge that the selected parameters are reasonable and representative.

2. The discussion section needs to be improved, and appropriate outlook should be supplemented.

The conclusion has been revised.

“During the study, the products obtained from composting of C. oleifera shell and goat dung for 76 days were in accordance with the Chinese organic fertilizer standard NY525-2012. In the composting process, TN, NO- 3-N, GI, Solvita maturity index increased with the composting process, while NH+ 4-N, C/N showed the opposite trend. According to the correlation analysis, TN, C/N ratio and GI can be used to evaluate the maturity of compost of C. oleifera shell and goat dung comprehensively. Among the four treatments, the longest duration of thermophilic phase was observed in A2 for 42 days (>50℃), with the highest total nutrient (5.75±0.11%) and lowest C/N value (15.83±1.10). It can be seen that the turning frequency of every 7 days can improve the quality of compost products. However, due to the diversity of raw materials and composting conditions, we need to do more research to acquire parameters with simple determination method and strong correlation with maturity to determine the appropriate turning frequency for composting.“

3. Are “sheep manure” and “goat dung” the same in meaning? If they are same, please unify the words.

The meaning we want to express is the same, but there are problems with the use of words. It has been revised in the article and unified as "goat dung".

4. The method of moisture determination should be introduced in the “materials and methods” section.

The method for determining moisture content is supplemented in this paper. “Moisture content was obtained by drying at 105℃ for 24h in a hot-air oven.”

5. Lines 145-146: why the temperature decreased to room temperature suddenly in all treatments?

It may be that heap dump causes a lot of heat loss, and the heat generated by microbial metabolism after a long time of heap dump is far from enough to replenish the lost heat, so the temperature suddenly drops to room temperature. Relevant expressions are supplemented in the article.

6. The resolution of the pictures in the manuscript is poor. 

Pictures have been replaced. The new picture was in 600 dpi.

---

## [Decision Letter · Decision Letter 1]

10 Sep 2019

[EXSCINDED]

Effects of turning frequency on the nutrients of Camellia oleifera shell co-compost with goat dung and evaluation of co-compost maturity

PONE-D-19-19121R1

Dear Dr. Zhang,

We are pleased to inform you that your manuscript has been judged scientifically suitable for publication and will be formally accepted for publication once it complies with all outstanding technical requirements.

With kind regards,

Sartaj Ahmad Bhat, Ph.D

Academic Editor

PLOS ONE

Additional Editor Comments (optional):

Reviewers' comments:

Reviewer's Responses to Questions

**Comments to the Author**

1. If the authors have adequately addressed your comments raised in a previous round of review and you feel that this manuscript is now acceptable for publication, you may indicate that here to bypass the “Comments to the Author” section, enter your conflict of interest statement in the “Confidential to Editor” section, and submit your "Accept" recommendation.

Reviewer #1: All comments have been addressed

Reviewer #2: All comments have been addressed

2. Is the manuscript technically sound, and do the data support the conclusions?

Reviewer #1: Yes

Reviewer #2: Yes

3. Has the statistical analysis been performed appropriately and rigorously? 

Reviewer #1: Yes

Reviewer #2: Yes

4. Have the authors made all data underlying the findings in their manuscript fully available?

Reviewer #1: Yes

Reviewer #2: Yes

5. Is the manuscript presented in an intelligible fashion and written in standard English?

Reviewer #1: Yes

Reviewer #2: Yes

6. Review Comments to the Author

Reviewer #1: Authors addressed all my concerns. Following the revisions, I think the manuscript has improved a lot. It can be accepted for publication.

Reviewer #2: (No Response)

7. PLOS authors have the option to publish the peer review history of their article (what does this mean?). If published, this will include your full peer review and any attached files.

Reviewer #1: No

Reviewer #2: No

---

## [Editor Report · Acceptance letter]

16 Sep 2019

PONE-D-19-19121R1 

Effects of turning frequency on the nutrients of *Camellia oleifera* shell co-compost with goat dung and evaluation of co-compost maturity 

Dear Dr. Zhang:

I am pleased to inform you that your manuscript has been deemed suitable for publication in PLOS ONE. Congratulations! Your manuscript is now with our production department. 

With kind regards,

on behalf of

Dr. Sartaj Ahmad Bhat 

Academic Editor

PLOS ONE